# Reversible thermal regulation for bifunctional dynamic control of gene expression in *Escherichia coli*

Xuan Wang[1,2,6], Jia-Ning Han [1,6], Xu Zhang[1], Yue-Yuan Ma[1], Yina Lin[1], Huan Wang[1], Dian-Jie Li[3], Tao-Ran Zheng[1], Fu-Qing Wu[1,4], Jian-Wen Ye[1,4,5✉] & Guo-Qiang Chen [1,2,4✉]

Genetically programmed circuits allowing bifunctional dynamic regulation of enzyme expression have far-reaching significances for various bio-manufactural purposes. However, building a bio-switch with a post log-phase response and reversibility during scale-up bio-processes is still a challenge in metabolic engineering due to the lack of robustness. Here, we report a robust thermosensitive bio-switch that enables stringent bidirectional control of gene expression over time and levels in living cells. Based on the bio-switch, we obtain tree ring-like colonies with spatially distributed patterns and transformer cells shifting among spherical-, rod- and fiber-shapes of the engineered *Escherichia coli*. Moreover, fed-batch fermentations of recombinant *E. coli* are conducted to obtain ordered assembly of tailor-made biopolymers polyhydroxyalkanoates including diblock- and random-copolymer, composed of 3-hydroxybutyrate and 4-hydroxybutyrate with controllable monomer molar fraction. This study demonstrates the possibility of well-organized, chemosynthesis-like block polymerization on a molecular scale by reprogrammed microbes, exemplifying the versatility of thermo-response control for various practical uses.

[1] Center for Synthetic and Systems Biology, School of Life Sciences, Tsinghua University, Beijing, China. [2] Tsinghua-Peking Center for Life Sciences, Beijing, China. [3] School of Physics, Peking University, Beijing, China. [4] MOE Key Lab of Industrial Biocatalysts, Department of Chemical Engineering, Tsinghua University, Beijing, China. [5] Center for Materials Synthetic Biology, CAS Key Laboratory of Quantitative Engineering Biology, Shenzhen Institute of Synthetic Biology, Shenzhen Institute of Advanced Technology, Chinese Academy of Sciences, Shenzhen, China. [6] These authors contributed equally: Xuan Wang, Jia-Ning Han. ✉email: jw.ye@siat.ac.cn; chengq@mail.tsinghua.edu.cn

Synthetic biology aiming to perform computational programs in living cells enables revolutionary developments in biotechnology[1,2]. Notably, scalable and robust gene circuits for uni- and bidirectional control of target genes over various timings and levels triggered by specific signals are desirable for diverse applications[3–5]. Generally, a dynamic control system of customized functions can be achieved by leveraging a variety of signal-sensing modules, including chemicals[6,7], intermediate metabolites[8–10], temperature[11–13], light[14–16], and cell population[17] responsive biosensors. In previous studies, many successful systems have been constructed for different purposes, such as enhanced production of various metabolic products including fuels[18], drugs[19], or other valuable chemicals[20] by microbes engineered using bidirectional dynamic control strategy.

More importantly, in contrast to the traditional uses for metabolic control of endogenous or heterogenous pathways as a "metabolic valves"[14,21], genetic programs of defined function demonstrate inconceivable capability of generating versatile patterns in narrowed down scales from colony size to molecular scales inspired by nature[22–24], for instance, programmable cell consortia and space-sensing coordinate patterning[2,25,26] based on quorum-sensing systems, block-like amyloid nano-fibers assembly using toggle switch[27], and so on.

However, to achieve dynamic optimization of cell factory engineering or nature-inspired patterning, there are still many challenges ahead of us to construct a well-designed circuit conferring exquisite, reversable and dynamic control for practical uses, which requires stringent and scalable control activity, instantaneous removal of sensing signal, fast-response availability in the post-log-phase, etc. Compared with the chemical- and cell density-response biosensing machinery used for dynamic metabolic control[28], temperature and light are attractive strategies with reversibility to address these challenges. In particular, thermosensitive genetic tools, which have been engineered for different uses, such as dynamic regulation for defining cell growth and production synthesis phases[11,12], gene therapy[29], etc., are more ideal for practical utilizations because of the convenient collocation, low cost, easy operability, and good dispersity of heat-transfer required in varied bioprocesses[30,31]. Therefore, engineering thermal-switchable bioswitch of bifunction can make possible the simultaneous activation and depression of distinct sets of genes in a temperature-dependent manner.

To address the interests, we demonstrate a versatile thermosensitive system, termed T-switch, for bifunctional dynamic control of gene expression in recombinant Escherichia coli, based on a temperature-associated transcriptional regulator CI857 [32]. Using the T-switch and its derivates, we first generated tunable and hierarchical tree ring-like patterned colonies under periodically temperature-changing circumstance, inspired by the natural tree ring formation responded to the seasonal variation of environmental temperature and humidity. Besides, we also obtained transformer cells among spherical, rod, and fiber shapes by controlling the expression of morphology-associated genes at different temperatures. Furthermore, T-switch was employed to modulate the biosynthesis of two building blocks of poly-hydroxyalkanoates (PHA)[33,34], enabling the chemosynthesis-like polymerization of diblock copolymer, poly(3-hydroxybutyrate)-block-poly(4-hydroxybutyrate), short as PHB-b-P4HB. Our results thus revealed the sophisticated gene expression control for narrowed down patterning, from tree ring-like colony on macroscopic scale, morphology-changable bacteria on microscopic scale, to ordered-assembled block biopolymers on molecular scale, which opens the possibility for ingenious tailor-made molecular assembly in vivo.

## Results

**Prototyping bifunctional dynamic control of T-switch.** In order to develop a temperature-dependent bifunctional bioswitch, namely, T-switch, of robust gene expression control in engineered E. coli, two cascaded modules were constructed based on a thermosensitive transcriptional regulator CI857 and a TetR-family repressor PhlF encoded, respectively, by gene cI857 and phlF[35] (Fig. 1a). The recombinant E. coli JM109SGL came from E. coli JM109SG with deficiency of sad and gabD gene reported in previous study, by an additional deletion of lacI (Supplementary Fig. 1)[36]. For the temperature sensory module (construct 165), the constitutive expression of cI857, a widely used mutant of cI from bacteriophage λ as a thermo-genetic tool[37], exhibits strong repression on $P_R$ promoter at 30 °C by forming dimmer CI857 complex to achieve active and inactive transcriptional control of repressor PhlF and reporter mRFP under 37 °C and 30 °C, respectively. To engineer bidirectional control function, construct 155 containing reporter sfGFP expression module controlled by $P_{PhlF}$, the corresponding promoter tightly inhibited by PhlF, was cascaded to receive the propagated signal of PhlF from temperature-sensing panel. Thus, the switchable bidirectional control mediated by temperature modulation between 30 and 37 °C can be characterized through the ON/OFF expression performance of sfGFP and mRFP (Fig. 1a).

To study the temperature-response function of T-switch from 30 to 37 °C, FACS analysis was carried out using recombinant cells of E. coli JM109SGL harboring constructs 155+165 grown in a 96-deep well plate for 12 h at different temperatures in Luria-Bertani (LB) medium. The control performance of construct group 155+165 is summarized in Fig. 1c, showing 35- and 1819-fold dynamic range of red fluorescence (mRFP) and green fluorescence (sfGFP), respectively, between 30 and 37 °C. Meanwhile, the performance of construct 155+165 cultured in the chemical defined minimal medium (M9) was tested, obtaining similar dynamic range in contrast to a LB medium (Supplementary Fig. 2). Interestingly, the low-temperature responsive ON-performance characterized by sfGFP panel exhibited a higher sensitivity and lower leakage in the OFF-stage (sfGFP under 37 °C vs. mRFP under 30 °C) compared to the high-temperature responsive panel characterized by mRFP (Fig. 1c). Significant decrease of leakage was observed with the addition of degradation tag AVV to the C-terminal of sfGFP and mRFP, respectively, resulting in a higher dynamic range except the performances at 30 °C (Supplementary Fig. 3). The topology-like performance of T-switch was studied by plotting the fluorescent intensity (FI) of sfGFP against mRFP (Fig. 1d), forming two separated regions A and B with dominant green and red fluorescence, respectively. Notably, these performances also conferred the simultaneous up- and downregulation, namely bifunctional control, of target genes by replacing sfgfp and mrfp at altering temperatures from 30 to 37 °C. Furthermore, the expression levels of two independent modules, including phlF-mrfp cluster from construct 165 and sfgfp (Fig. 1b), were characterized for linear regression analysis with $R^2 = 0.9868$ and slope $k = 1.065$ (Fig. 1e), indicating negligible expression variance among different target gene clusters controlled by temperature sensing panel derived from construct 165. In addition, considering the compatibility of T-switch circuit functioning in the start host at 30 and 37 °C, transcriptome profiling was performed to study the effects of genome-wide mRNA levels. Results showed highly consistent total mRNA abundance and distribution under different conditions except 250 genes that were significantly downregulated at 37 °C, probably due to the minor crosstalk of PhlF expression in cellular regulatory network (Supplementary Figs. 4 and 5). Generally, a stringent and robust T-Switch had been successfully constructed to achieve bifunctional regulation

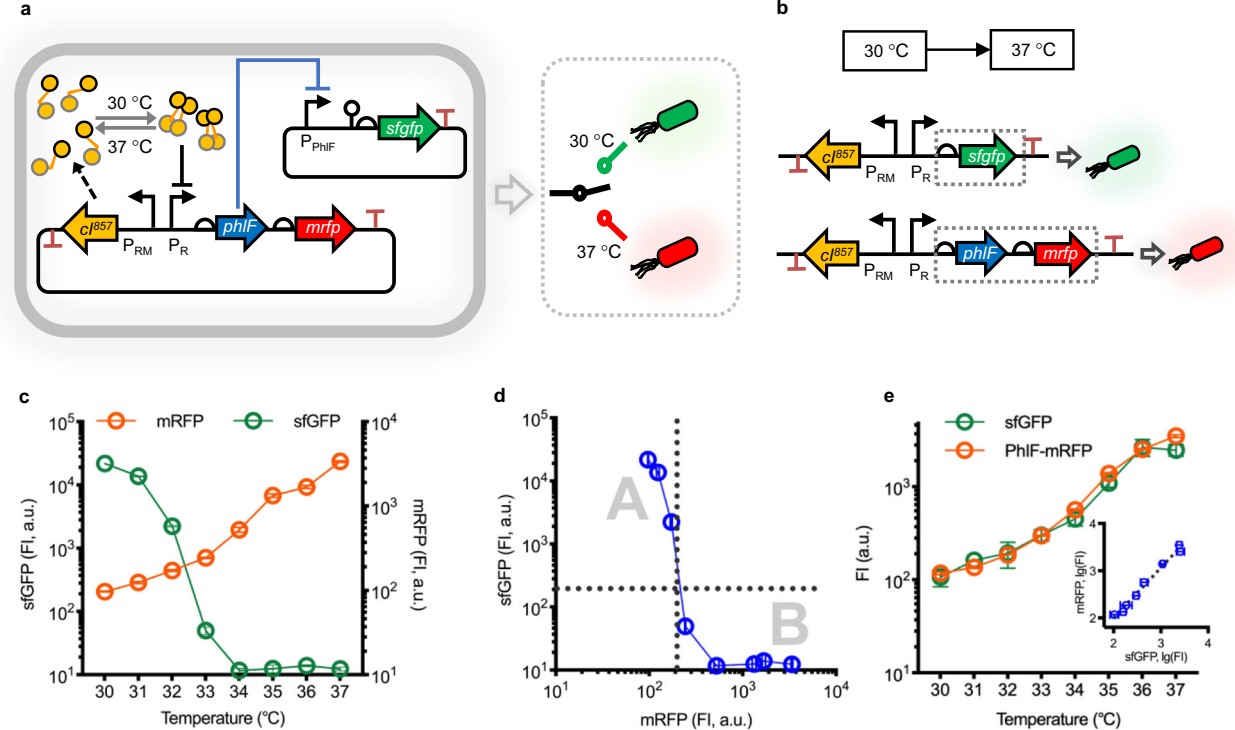

**Fig. 1 Design and characterization of thermosensitive bifunctional bio-switch (T-switch). a** Circuits of T-switch using a NOT gate design enables bifunctional gene expression, of which the input is the repressor PhlF related to a temperature-controlled system based on $cl^{857}$ ($cl$), and the output is a reporter *sfgfp* encoding sfGFP under the promoter $P_{phlF}$. Another output is a reporter *mrfp* encoding mRFP placed downstream the *phlF* gene (*phlF-mrfp*) for achieving bifunctional control of gene expression. **b** Promoter activity and orthogonality of the $CI^{857}$-regulated panel are characterized based on two expression cassettes: *sfgfp* as a reporter alone in **a** and *phlF-mrfp* cluster. **c** Data generated using the constructs from part **a** via cytometry analysis are used to characterize the bidirectional temperature-response functions of the T-switch. **d** Data gathered onto a single transfer function between two reporters, namely, sfGFP and mRFP, are identical to those shown in c from the same experiments. Each point represents a condition of cultural temperature from 30 °C to 37 °C. **e** The fluorescence intensity was measured from each temperature from 30 °C to 37 °C for controlling the expression levels of *sfgfp* and *phlF-mrfp*. The fluorescence ratios between sfGFP and mRFP expression from each temperature point collapse onto a single function for linear regression analysis with $R^2 = 0.9868$ and slope close to 1 (1.065), which was plotted in log-log coordinates ($\log_{10}$). Data are presented as mean ± s.d. of three replicates. FI, Fluorescence Intensity in arbitrary unit (a.u.).

of gene expression with a significant tunable dynamic fold-change.

**Optimization and characteristics of T-switch.** On the basis of the original T-switch, constructs 155+165, further bioswitch engineering was developed by introducing negative feedback loop control of the temperature sensing panel (Fig. 2a). Specifically, repressor LacI, encoded by *lacI* under $P_{PhlF}$ promoter, associated operator LacO, was introduced to achieve a negative feedback control of the transcriptional activity of $P_R$ promoter under low-temperature (30 °C) responded sfGFP-ON stage. First, the seed cultures of recombinant cells harboring constructs 155+165 and 147+167, respectively, were prepared at 30 °C, then inoculated and grown at 37 °C for 12 h, followed by the time course monitoring of fluorescence intensity of both reporters. The static temperature-response performances of two T-switch combinations were thus characterized. Constructs 147+167 exhibited a tighter sfGFP-OFF control with a lower leakage and lag-activation of mRFP-ON control compared to the original constructs 155+165 (black triangle shown in Fig. 2b).

More combinatory designs were constructed with the addition of degradation tags, AAV and LVA (sequences are shown in Supplementary Table 2), to the C-terminal of sfGFP and mRFP, respectively, to study the dynamic temperature-response performances during different growth phases at altering temperatures from 30 to 37 °C in 0–12 h (at 0, 2, 4, 6, 8, 10, and 12 h) after

inoculation (Fig. 2c and Supplementary Fig. 6). In contrast, cell cultures maintained at 30 °C for 24-h growth after inoculation were used as control group labeled with "−." Results showed that the mRFP-ON response controlled by the temperature sensing panel was still active before 10 h with less than 50% decrease of fluorescence intensity. However, the sfGFP-OFF control could be activated in the first 6 h, approximately in the mid-log phase (Fig. 2c and Supplementary Fig. 13a). Constructs 147+168 exhibited a poor mRFP-ON performance with discrete distribution of fluorescence due to the joint weakening of LacO and AAV tag (Supplementary Fig. 7). In addition, correlations of four *phlF-mrfp* modules with combinatory supplementation of LacO operator and AAV tag controlled by the temperature-sensing panel were characterized. They had highly pairwise linear correlations with $R^2$ (square of Pearson correlation coefficient) higher than 0.93 (Supplementary Fig. 8). Interestingly, the insertion of LacO resulted in a nearly five-fold increase of output level, including the leakage under OFF-stage at 30 °C. On the contrary, the addition of AAV tag displayed about 100-fold decrease of basal leakage.

However, the reversible control of bifunction from 37 to 30 °C cannot be achieved possibly due to the durable tight control of PhlF. Thus, time-lapse photography was applied to study the dynamic response from 37 to 30 °C right after inoculation in a single-cell level leveraging confocal online monitoring with recombinant cells incubated underneath a layer of solid 1.0% LB agarose (~1.5 mm) (Fig. 2d). Under this situation, the

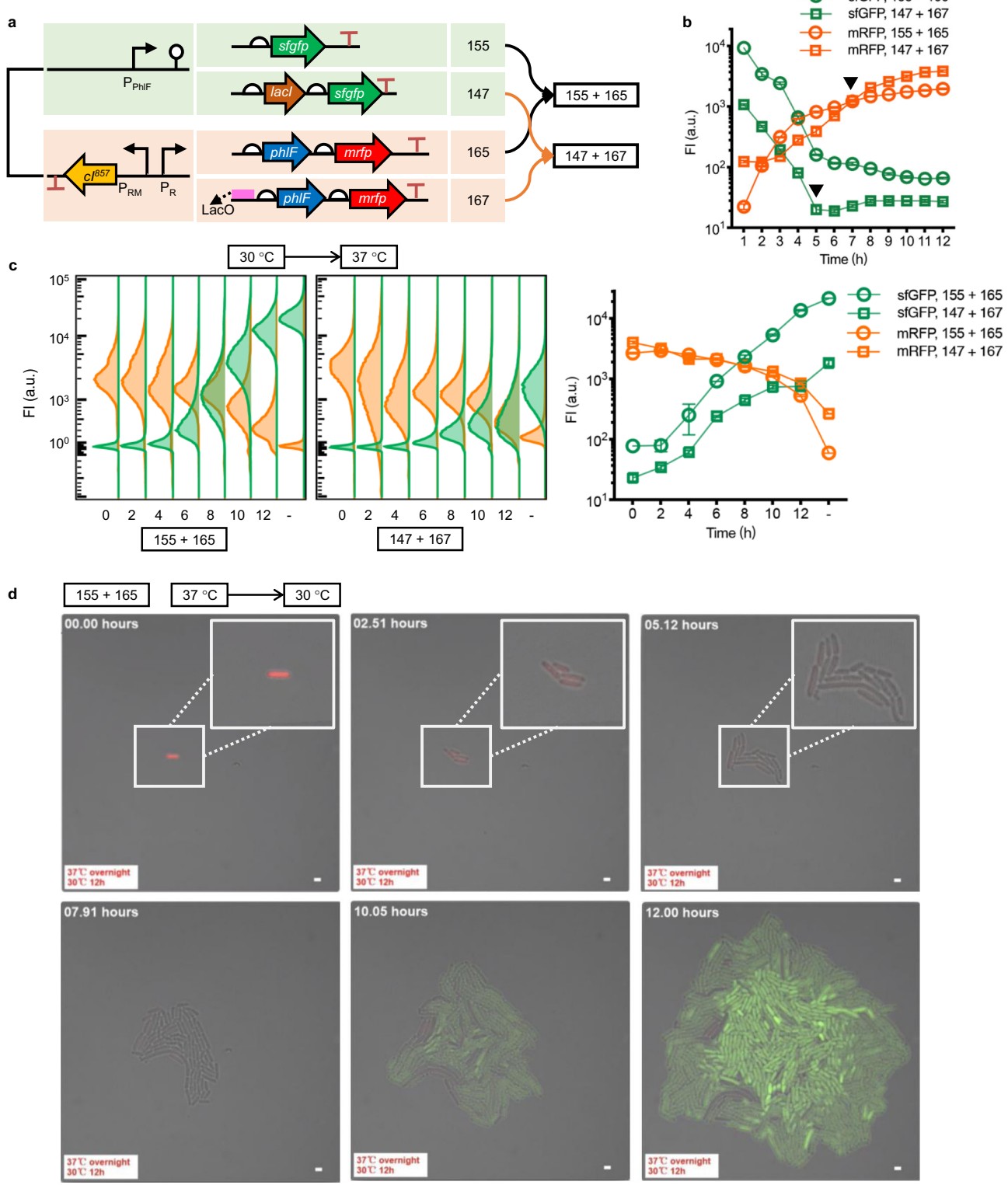

**Fig. 2 Switch-testing of bifunctional on- and off-responses. a** Constructs based on T-switch, the 155+165 group, by introducing negative feedback loop control of input signals, namely 147+167 group. *lacI* gene was co-expressed with *sfgfp* to stringently repress the promoter activity of $P_R$ with the downstream insertion of a LacI-associated operator, LacO. **b** Fluorescence intensity (FI) monitoring of time course of T-switch circuits by altering temperature from 30 °C to 37 °C right after inoculation to characterize the response-curve under thermal inductions. Cells were grown at 30 °C throughout the pre-cultivation from a single colony. **c** On- and off-response performance of T-switch constructs shown in **a** in different growth phases. Recombinant cells were grown for 12 h after changing temperature from 30 °C to 37 °C. Left: distribution and variance of fluorescence, including sfGFP and mRFP, monitored using a flow cytometer; right: quantitative comparisons of on/ off performances. Data in **b** and **c** are presented as mean ± s.d. of three replicates. **d** 12 h Time-lapse photography of the temperature-responsive performances of engineered E. coli harboring T-switch, and constructs 155+165, during a shift from 37 °C to 30 °C. Samples comprising 1 μL of the cell suspension were injected and incubated underneath a layer of solid LB with 1.0% agarose (~1.5 mm) containing the relevant antibiotics at 30 °C from overnight cultures at 37 °C. Since the tightness and ultra-sensitivity of *tetR* family repressor PhlF, the activation of sfGFP lagged behind the quenching of mRFP by at least 4 h.

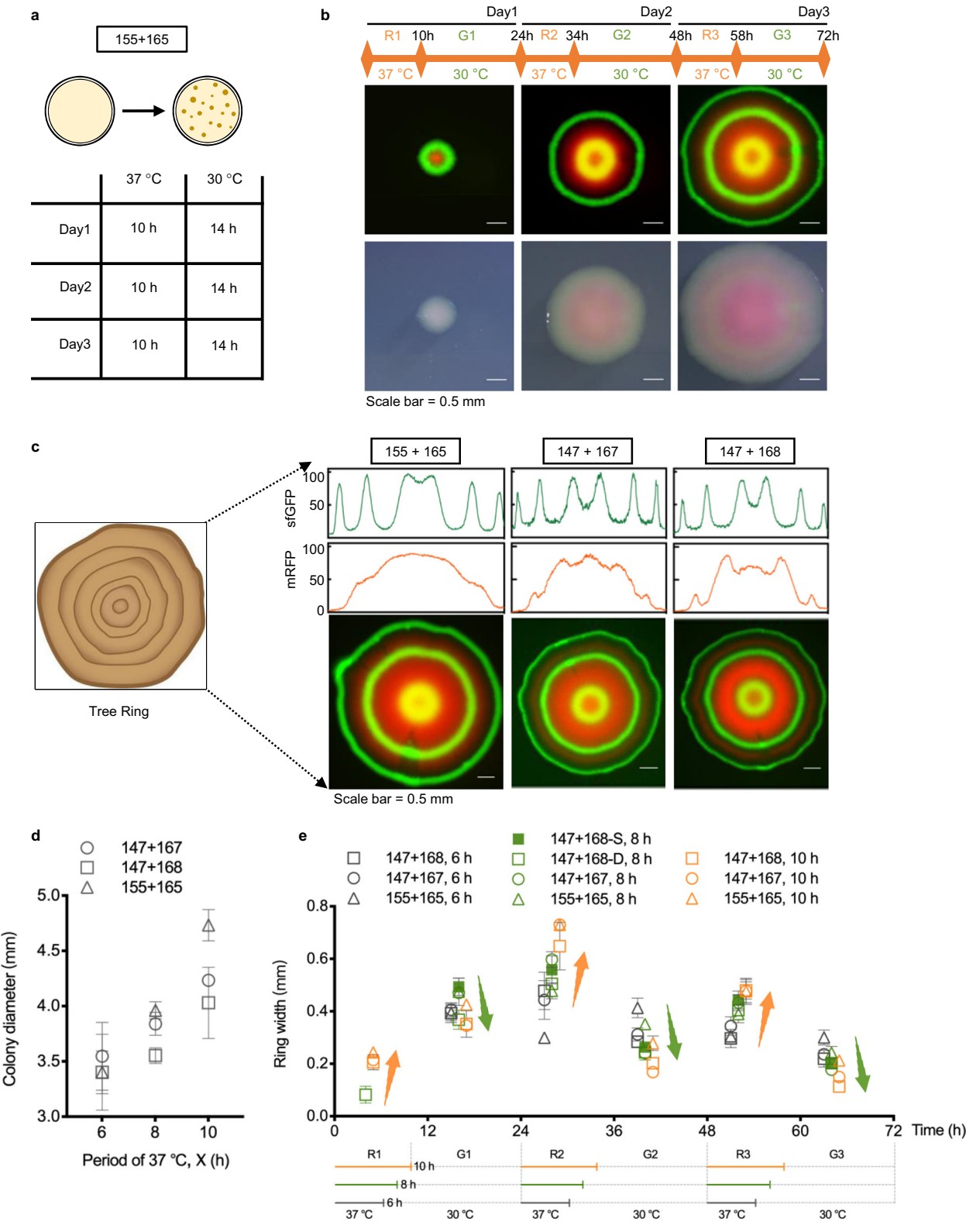

Scale bar = 0.5 mm

dynamic regulation of sfGFP-ON and mRFP-OFF could be obtained. However, at least 4-h time-lag of sfGFP-ON activation was observed after the entire quenching of mRFP (Supplementary File 1), demonstrating that long hours of degradation and multiple generations of cell growth dilution of PhlF are necessary to generate effective dynamic control of bifunction from 37 to 30 °C.

**Tree ring-like patterning by temperature-response grown colonies.** Artificial genetic circuits are powerful tools to explore the fundamental insights of hierarchical patterns of nature. To illustrate the feasibility of T-switch designs, we tested their periodic thermo-response performances of bifunctional control during colony formation by mimicking the formation of natural tree ring under the seasonal variation of temperature and humidity.

**Fig. 3 Tunable formation of tree ring-like colonies. a** Three-day tree ring-like colony formation by thermally controlled expression of sfGFP and mRFP. Specifically, plates spread with recombinant cells from cell cultures at 37 °C were incubated in periodical changing circumstances at 37 °C and 30 °C for 10 h and 14 h, respectively. **b** The colony growth image was recorded every 24 h to study the forming process of rings in red or green. The band width of outer rings became more and more narrow due to the gradually saturated growth of colony, especially for green rings compared to that of the red ones. **c** Growth of a single colony harboring various T-switch circuits, including 155+165 from part **b**, 147+167 and 147+168 groups, by periodically changing temperature between 30 °C and 37 °C. Construct of 168 was derived from 167 by adding AAV degradation tag to reporter *mrfp* to reduce the memory effects of thermal controlled system. Fluorescence intensity (FI) was measured by imageJ from images and normalized by the maximum value of sfGFP and mRFP measured from every single image, ranging from 0 to 100. **d** Diameter of tree ring-like colonies increased with longer temperature cycles at 30 °C. **e** Ring-width of sfGFP- and mRFP-ring were obviously affected by different T-switch designs and temperature cycles from 30 °C to 37 °C. Hollow and solid squares represent the visible and missing formation of the first red ring (R1) generated by constructs 147+167, respectively, namely 147+167-D (2 colonies) and 147+167-S (4 colonies), respectively. Data in **d** and **e** are presented as mean ± s.d. of at least 2 collected colonies. *p* values of **d** and **e** are displayed in Supplementary Fig. 12.

Colonies of recombinant cells showed spatially distributed double-color ring patterns of mRFP (red) and sfGFP (green). First, recombinant cells harboring the start constructs 155+165 were spread on 10LB agar plate for 72-h growth by changing the temperature periodically, with 10-h growth first at 37 °C then 14-h at 30 °C per day as a cycle (Fig. 3a). The colonies with fluorescence rings were photographed under bright- and dark-field microcopy at the end of each cycle (Fig. 3b). Interestingly, the inner green ring (G1) turned yellow in the next two cycles of incubation due to the integration of green (sfGFP) and red (mRFP) colors. Notably, the sfGFP rings were distinguishable, while the mRFP FI was stepwise enhanced from outer- to inner-region, because the in situ memorial expression of mRFP was periodically activated and stacked once the colony was exposed at 37 °C (see cartoon video in Supplementary File 2). To form the distinct red rings, the OFF-stage of mRFP should be steadily retained whenever the temperature changed from 30 to 37 °C.

Therefore, two more T-switch designs, constructs 147+167 and 147+168, were joined to generate rigorous colony ring pattern with stringent control of mRFP and sfGFP by introducing negative feedback control from LacI and degradation tag AAV (Fig. 3c). Normalized FI by ImageJ was used to characterize the expression-level variation of sfGFP and mRFP involved in different color rings. More specifically, the color formation of inner green ring (G1) was significantly improved with reduced integration effect of red color, resulting from fluctuant radical distribution of normalized mRFP FI compared to the stepwise tendency of constructs 155+165. Accordingly, typical peaks of normalized green fluorescence appeared in the valley of the red ones, indicating an optimized staggered formation of red and green rings with distinctive boundaries. Furthermore, characteristics of incubation time at 37 °C ($X = 6$, 8, and 10 h) of recombinant cells in each cycle were performed to achieve tunable pattern formation (Supplementary Figs. 9–11). Statistically, longer incubation time ($X$) at 37 °C generally leads to formation of a larger size colony with increased red ring width (R1, R2, and R3) and decreased green ring width (G1, G2, and G3) (Fig. 3d, e and Supplementary Fig. 12), which is possibly relevant to the growth phase of colony formation. The protein expression activity decreased while the colony growth reached the stationary phase in the third cycle incubation, which also led to decreased fluorescent peak values from the inner to outer color rings. Notably, the inner red ring (R1) was drowned in the inner green ring (G1) because of the short-term (6 and 8 h) growth at 37 °C except two of the six collected colonies from constructs 147+168 exhibited recognizable double inner rings of R1 and G1 (hollow square in green in Fig. 3d, termed 147+168-D, 8 h). In conclusion, the tunable formation of tree ring-like colony pattern by engineered *E. coli* carrying various T-switch designs demonstrated promising thermal control capability of bifunction for further application trial in functional gene expression control.

**Thermal regulation of cell morphology.** Dynamic morphology control over time and levels promote the process of understanding fundamental morphological insights as well as shaping engineered cells in industry[38–40]. Here, we engineered a prototype T-switch (constructs 145+221, termed TM-switch) to modulate the expression of morphology-associated genes instead of reporters, including cell skeleton gene *mreB* and cell division gene *ftsZ*, respectively, related to the formation of spherical and fiber shapes (Fig. 4a). First, static test was performed to culture the recombinant cells at 30 °C, 33 °C, and 35 °C to form spherical, rod, and fiber shapes, respectively (Fig. 4b). Particularly, because of the low expression level of T-switch at 33 °C with a green FI value close to 200 in bifunctional control panels (Fig. 1d), cells harboring TM-switch were maintained in normal rod shape compared to the heteromorphic phenotype at 30 and 35 °C. Quantitative cell length measurement of at least 150 cells from confocal images from each group revealed remarkable increasing trend along with the activation of *ftsZ* and inactivation of *mreB* gene from 30 to 35 °C (Fig. 4c and Supplementary Files 3–5).

To further explore dynamic morphology controls, recombinant cells carrying TM-switch were cultured in a chamber within a microfluidic device for online recording of cell growth with thermo-response that shapes variation under different control strategies (Fig. 4d), including single-step control from 30 to 35 °C (left panel), from 33 to 35 °C (middle panel), and dual-step control from 30 to 33 to 35 °C (right panel, Supplementary File 8). Specifically, for single-step control groups, overnight seed cultures at 30 and 33 °C were injected in chip chambers for 8 h growth at 35 °C in flowing fresh LB medium, while dual-step control group was able to maintain the cell growth at 30 °C for 1.5 h, the same for the seed cultures, before changing the temperature to 33 and 35 °C orderly after 1.5- and 4-h growth, respectively. Video captured at 1, 3, and 5 h of single-step control groups exhibited remarkable cell elongations from spherical and rod stage (Supplementary Files 6 and 7). However, minor elongation of cells was observed at 4 and 6 h captured images with temperature changing from 33 to 35 °C from dual-step control group, probably due to the saturated cell population since longer cells were easier to be observed in neighbor chambers with less cell counts (Supplementary File 8). This study not only illuminates the thermo-response bifunctional control of target genes of functions, but also hints at the practical uses of efficient output in timing control of temperature alteration.

**Ordered assembly of block-copolymer poly(3-hydroxybutyrate)-block-poly(4-hydroxybutyrate).** Engineering dynamic control circuit of mono- or bifunction to modulate target pathways for enhanced bioproduct accumulation is a commonly used strategy to make effective production by decoupling the cell growth phase and production phase[41–43]. However, the de novo ordered assembly of biopolymers on a molecular scale, which requires

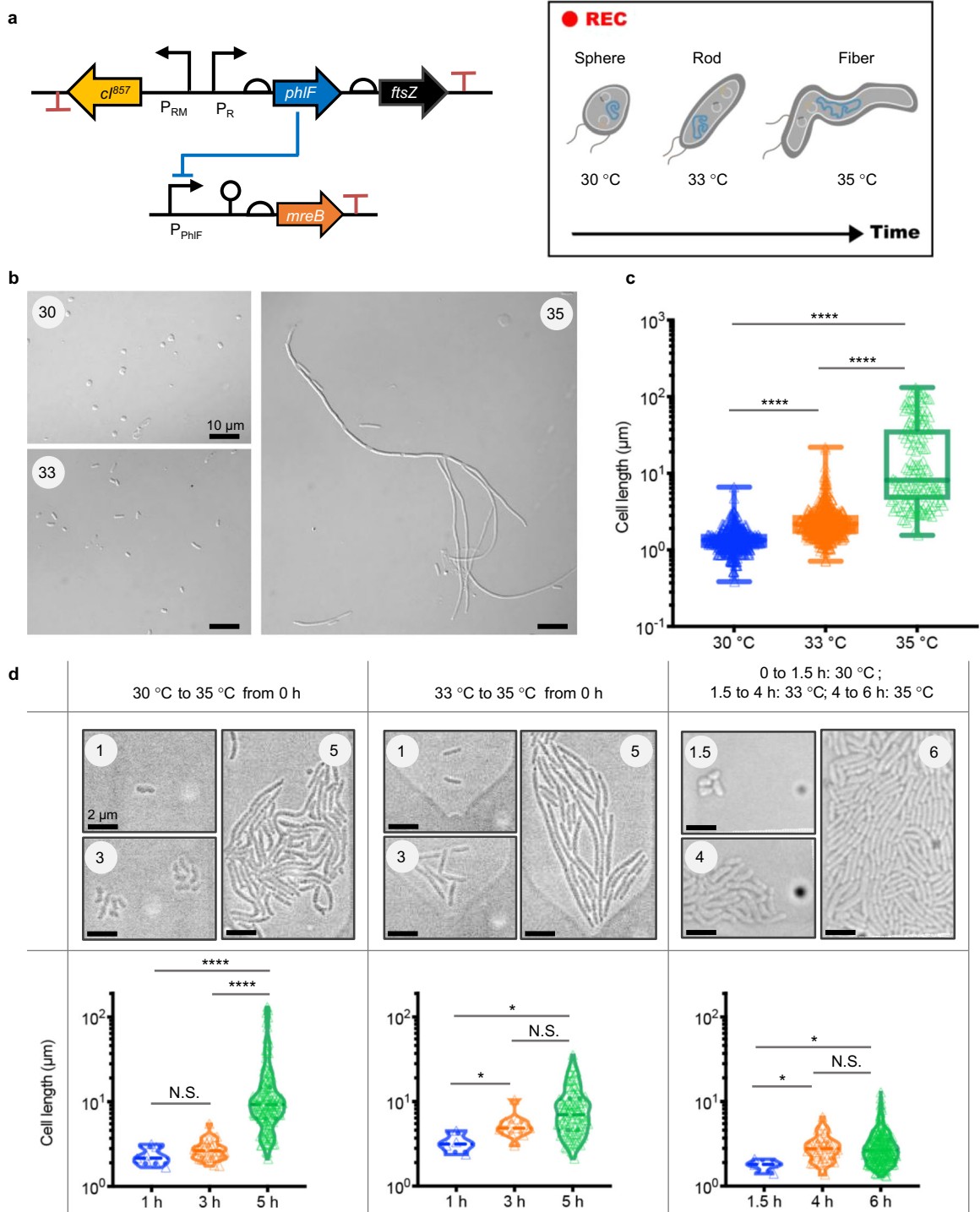

**Fig. 4 Thermal responsive cell morphologies changes from rod to spheres or to fibers. a** Construct for controlling cell morphology. The expression of two morphology related genes, *mreB* and *ftsZ*, enabling formations of sphere- and fiber-shape cells, respectively, were controlled by the T-switch system. **b** Confocal imaging of cells harboring 145+221 constructs cultured at constant temperature, 30 °C, 33 °C and 35 °C, scale bar = 10 μm. **c** Quantitative measurements of cell lengths by imageJ from at least 9 images from part **b** containing over 150 captured cells manually. **d** Bifunctional dynamic control of cell morphology among shapes of spheres (*mreB* overexpressing), rods (normal cell type) and fibers (*ftsZ* overexpressing) were on-line recorded in a microfluidics with a scale bar of 2 μm. Sample sizes of collected cells of each time point varied significantly depending on the growth phase, 3–10 cells at 1 h, 10–40 cells at 3 h, and 100–240 cells at 5 h. All data in **c** and **d** are displayed in Box-plot: a value of median, quantiles, mini- and maxi-mum. One-way ANOVA with Tukey-Kramer test was used in **c** and **d**. *p* value: N.S. not significant; \**p* < 0.0332; \*\**p* < 0.0021; \*\*\**p* < 0.0002; and \*\*\*\**p* < 0.0001.

exquisite expression control over time and levels with well-organized collaborative response, is still challenging to achieve comparable performance of chemosynthesis polymerization. To our knowledge, PHA is a family of microbially synthesized

biodegradable[44] biopolymers with tunable thermo-mechanical properties depending on the components, molecular weights, and polymerization manners including homopolymer, random-, and block-copolymer (Supplementary Fig. 14)[33]. Therefore, in

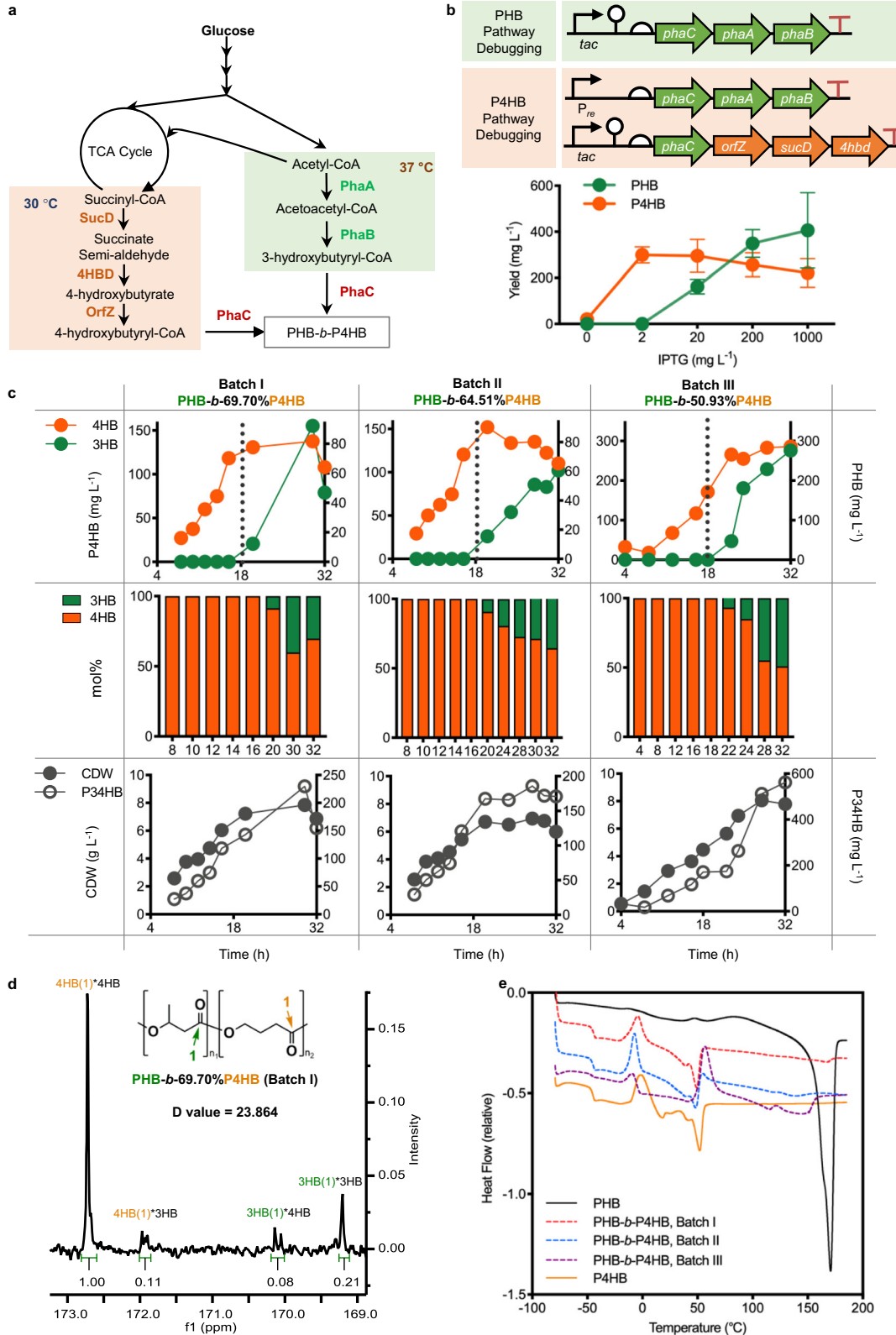

addition to the successes of tree ring-like colony and transformer cell, herein, further efforts were carried out to expand the T-switch into more difficult uses of in vivo ordered polymerization of diblock PHA composed of poly (3-hydroxybutyrate) (PHB) and poly (4-hydroxybutyrate) (P4HB) blocks, namely PHB-*b*-P4HB (Fig. 5a).

Before the circuit construction, two major tests including cell growth characterization in various scales and transcription-response sensitivity of target pathways were implemented to determine the operational time range for temperature alteration and pathway control strategy, respectively, with the "Begin with the end in mind" rational design (Supplementary Figs. 14 and 15).

**Fig. 5 Synthesis of block copolymer PHB-*b*-P4HB by recombinant *E. coli*. a** PHB-*b*-P4HB synthesis pathway from glucose as a sole carbon source, which is grouped into three modules including syntheses of 3-hydroxybutyrate-CoA synthesis (highlighted in light green, activated at 37 °C), 4-hydroxubutyrate-CoA (highlighted in light orange, activated at 30 °C) and constitutive expression of PHA synthase PhaC encoded by *phaC* controlled by *porin* promoter, respectively. **b** Debugging the expression sensitivity of 3-hydroxybutyrate-CoA and 3-hydroxybutyrate-CoA synthesis pathways under the control of *tac* promoter induced by various concentrations of IPTG, respectively. 4HB synthesis pathway has more active performance in a low expression level with saturated P4HB accumulation in the presence of over 2 mg L-1 IPTG. Error bars, mean ± s.d. of three replicates. **c** Fed-batch fermentative production of PHB-*b*-P4HB was carried out for 3 times utilizing the same bioprocess under changing temperatures from 30 °C to 37 °C after 18 h growth in a 7 L bioreactor. Upper panel displays the accumulations of block PHB and P4HB in PHB-*b*-P4HB during the bioprocessing; middle panel shows 4HB molar fractions during the formation of PHB-*b*-P4HB; bottom panel displays the accumulative tendency of cell dry weights and PHA content of time course. **d** D-value of PHB-*b*-P4HB production from Batch-I determined by NMR C$^{13}$. *x*-axis: chemical shift (ppm); *y*-axis: intensity of chemical shift. **e** Thermodynamics assays of PHB-*b*-P4HB from three repeated batches including melting temperature ($T_m$) and glass transfer temperature ($T_g$), using differential scanning calorimetry (DSC). PHB from Sigma and P4HB from Tepha were used as control for comparative analysis. *x*-axis: temperature range of cooling-heating cycles; *y*-axis: relative heat flow rate generated from DSC scanning.

Notably, two heterogenous pathways, including *phaA-phaB* encoding β-ketothiolase (PhaA) and NADH-dependent acetoacetyl-CoA reductase (PhaB)[36], and *orfZ-sucD-4hbd* encoding CoA transferase (OrfZ), succinate semi-aldehyde dehydrogenase (SucD), and 4-hydroxybutyrate dehydrogenase (4HBD)[45], involved in the synthesis of 3HB-CoA and 4HB-CoA, the polymerization precursors for PHB-*b*-P4HB mediated by PHA synthase encoded by *phaC*, were constructed in the start host *E. coli* JM109SGL independently under the control of *tac* promoter. These constructs allow us to study their transcriptional levels for both PHB and P4HB synthesis induced by the same dosage of IPTG (Fig. 5b). Remarkably, the constitutive co-expression of PHB synthesis cluster from *Cupriavidus necator* (namely *Ralstonia eutropha* H16) was employed to reduce the toxicity of 4HB-CoA synthesis pathway for normal cell growth[46]. Results showed that the 4HB-CoA synthesis pathway exhibited lower expression-level limit with saturated activity in 4HB-CoA synthesis in the presence of 2 mg L$^{-1}$ IPTG compared to the one of 3HB-CoA synthesis (Fig. 5b and Supplementary Fig. 15c, d).

Based on these pre-debugging knowledge, the control strategy for PHB-*b*-P4HB synthesis was developed using constructs 169+170 carrying 4HB-CoA synthesis cluster controlled by P$_{PhlF}$ with tight-control leakage (activated at 30 °C) and 3HB-CoA synthesis cluster controlled by temperature-sensing panel with acceptable leakage (activated at 37 °C), while the *phaC* gene was constitutively expressed using *porin* promoter for continuous polymerization activity (Fig. 5a and Supplementary Fig. 16a). In addition, two degradation tags, LVA and AAV, were added to the C-terminal of PhaA and PhaB, respectively, to reduce the leakage under P$_R$ promoter. Subsequently, a pre-culture test was conducted in 1-L quadruple fermentors (Infors, Switzerland) by alternating temperature from 30 to 37 °C at 9, 12, 15, and 18 h, respectively, during 32 h of cultivation to identify the optimal time moment for PHB-*b*-P4HB synthesis (Supplementary Fig. 16b, c).

Generally, a diblock copolymer should have a D-value greater than 10 calculated from C$^{13}$ NMR spectrum. Meanwhile, two characteristic peaks of melting points ($T_m$) resulted from the diblock regions can be detected by differential scanning calorimetry (DSC) profiling[47]. Interestingly, prolonging the cultural time at 30 °C could significantly increase the 4HB molar fraction and D-value of PHB-*b*-P4HB block copolymer but with noticeable decline of cell mass and product titer because of the long-term activation of toxic 4HB-CoA synthesis pathway (Supplementary Figs. 16 and 17). Among these four fed-batch fermentations, the temperature alteration time at 18 h from 30 to 37 °C displayed dominant diblock features, having a D-value over 23 (Supplementary Fig. 17d), and two distinguished $T_m$ peaks in DSC profiling, which represented the typical melting points ($T_m$) of homopolymer of PHB and P4HB in the diblock copolymer (Supplementary Fig. 18).

Subsequently, three independent trials of fed-batch cultivations were conducted in a 7-L bioreactor (Thermal, USA) to obtain significant amount of PHB-*b*-P4HB containing 69.7% (Batch-I), 64.5% (Batch-II), and 50.9% (Batch-III) molar fraction of P4HB with D-values larger or close to 10 (Fig. 5c, d and Supplementary Fig. 19). Based on the time course recording of PHB and P4HB contents, the switching performance of ON- and OFF-control of PHB and P4HB synthesis were tightly and smoothly manipulated by changing temperatures from 30 to 37 °C at 18 h of growth (dash line) with undetected leakage of PHB accumulation (green dots) before 18 h, and effective termination of P4HB synthesis (orange dots) after 18 h compared to the gradually climbing of cell dry weight and PHB-*b*-P4HB titer (Fig. 5c).

DSC profiling of three-batch PHB-*b*-P4HB (dash line) also showed two characteristic $T_m$ peaks compared to PHB (solid line in yellow) and P4HB (solid line in black) (Fig. 5e), of which PHB-*b*-P4HB from Batch-I exhibited the best performance with dominant $T_m$ peaks and D-value close to 24 (Fig. 5d). Interestingly, by leveraging the leakage nature of temperature-sensing panel and ultra-sensitivity of 4HB-CoA synthesis cluster expression, further attempts of exchanging the control panels of 3HB-CoA and 4HB-CoA synthesis pathways (constructs 149+150) resulted in synthesis of random copolymers P(3HB-*co*-4HB) with increased 4HB molar fraction from 30 to 89% when maintaining the culture temperature at 30, 31, and 32 °C throughout the 32-h fermentation bioprocess (Supplementary Fig. 20). In contrast to the PHB-*b*-P4HB fermentation process, the synergetic accumulation pattern of PHB and P4HB represents the simultaneous polymerization of 3HB and 4HB monomers involved in random P(3HB-*co*-4HB) synthesis, leading to D-values close to 1 (Supplementary Fig. 21), and single or undetected $T_m$ peaks in DSC profiling (Supplementary Fig. 22). Otherwise, the mechanical properties of PHB-*b*-69.7% P4HB (Batch-I in Fig. 5c) displayed remarkable improvements of Young's modulus (23-fold) and tensile strength (2.3-fold) compared to the copolymer formed with a similar 4HB molar fraction of P(3HB-*co*-64%4HB) (batch at 31 °C in Supplementary Fig. 20b) (Supplementary Table 3).

## Discussion

Generally, a thermal switch has its advantages for scale-up microbial fermentations due to the uniform dispersity and easy operability of temperature control with negligible bias based on the proven industrial heat- and mass-transfer control. Therefore, robust temperature-simulated gene expression could be achieved even in a post-log-phase of cell growth (Supplementary Figs. 6 and 13), which suggests that thermosensitive tools are available in different cultural scales with strong response performances. Meanwhile, temperature-induced bidirectional control for diblock copolymer synthesis were successfully conducted in fed-

batch fermentations in a 1 and 7-L bioreactor, respectively, of which the $OD_{600}$ reached close to 50 for temperature alteration (Supplementary File 10), making the strong case for demonstrating the potential uses of thermal switch in the high cell density growth for larger scale fermentation processes. In addition, the removal of an input signal mediated by the temperature alteration confers the reversable and even periodical control of target genes manually for different purposes, especially for bio-inspired patterning, such as tree ring-like colonies obtained in this study. However, the cells harboring construct 155+165 grown in colony and liquid culture, especially for chemo-state culture with strong and sustainable dilution effects of cell division, exhibit totally different activities, because the recombinant cells grown on plate are spatially immobilized without free division activity but still can sense and response to the signal of temperature changes. For example, cells in the earlier rings formed at 30 °C are able to express mRFP in the later growth at 37 °C (Supplementary File 2), which requires careful design for problem-based complementation, such as construct 147+168 with the introduction of a negative feedback control. As a result, the enlarged transition regions of the green and red rings were observed after the introduction of two repression systems PhlF-PhlO and LacI-LacO (construct 147+168), since the recovered expression of related reporter highly depends on the radial dilution effect of cell division, forming a non-color gap between two neighbor rings. Generally, the compatibility of artificially designed T-switch in recombinant cell systems could be further improved by substituting the TetR-family repressor, PhlF, to avoid the non-objected repression of endogenous gene expression (Supplementary Fig. 5c). Otherwise, engineering thermal-associated regulators to change the operational range of temperature are a promising approach to achieve multiple-functional control[29], including various growth rates, morphology, and products. Overall, our T-switch circuits offer an easily established and high-performing solution for bidirectional dynamic regulation of gene expression based on currently used fermentation systems.

The in vivo ordered assembly of biomacromolecules[48–51] or nano complex, such as block-like amyloid fibers[27], is an interesting but challenged topic in synthetic biology, which required stringent and collocative control of well-characterized target gene sets. It was reported that the block co-polyesters, including PHA[52], PLGA[53], and other kinds of block copolymer[54], fabricated in chemosynthesis exhibit enhanced performances for material processing and medical uses. Besides, more attempts of various tailor-made biopolymer fabrications have been studied for high value-added applications[55–57]. Here, we used T-switch to dynamically manipulate the expression of two independent pathways for 3HB-CoA and 4HB-CoA synthesis and achieved the de novo synthesis of diblock copolymer PHB-b-P4HB with remarkable improvements on mechanical properties. This study opens a possibility of fabrication of diverse block co-polyesters or nano complex using the sustainable and renewable microbial cell factories that rival the chemical catalysis processes. However, it is important to note that the poor cell growth probably due to the intrinsic toxicity of 4HB-CoA synthesis pathway could be improved using static optimization approach, which has been previously proven in *Halomonas spp.*[36].

In summary, the thermal switch based on temperature-dependent regulatory proteins enables a powerful, applicable and scalable strategy to modulate the expression of gene sets for dynamic control of cellular metabolism[11,12]. This study has demonstrated that a thermal switch of bifunction, termed T-switch, holds promise in cell morphology control, well-organized colony pattern, and in vivo ordered assembly of biomacromolecules in a chemosynthesis-like manner combined with metabolic engineering. These successes put programmable gene circuits into desirable and diverse functions efficiently triggered by temperature as a single input, as well as define a rational pipeline for the construction and utilization of gene circuits in living cells with well-prepared debugging measures using "Begin with the end in mind" design.

## Methods

**Strains, plasmids, and media**. The chemical competent cells of *E. coli* DH5α (Biomed, China) were used for plasmids construction in this study. And *E. coli* JM109SGL, namely *E. coli* JM109SGL derived from *E. coli* JM109SG by deleting *lacI* gene (see Supplementary Methods and Supplementary Fig. 1), was used as a start host for various assays, including circuit characterization by FACS, single-cell study on microfluids devices and agar plates, and PHA copolymer production, respectively, unless specifically noted. In particular, *E. coli* JM109SG was previously constructed by Li et al. by knocking out *sad* and *gabD* genes to block the effluxes of succinyl semi-aldehyde involved in P4HB synthesis from glucose only, and thus enhancing the molar fraction of 4HB component in P(3HB-*co*-4HB) production[36]. All plasmids were constructed by Gibson Assembly toolkits (NEB, USA) based on two expression vectors containing p15A origin of replication and *Cm* resistance, named p15a vector, and pSC101 origin of replication and kanamycin (*Kan*) resistance, named pSB4K5 vector, respectively. Designs and detailed information of these two mother-plasmids and their derivates are listed in Supplementary Table 1 and Supplementary File 9, respectively. Genetic design and sequence reading were performed by SnapGene v3.2.1. Primers used in this study are listed in Supplementary File 9. Target clones with correct sequences by Sanger sequencing (Ruibiotech, China) were extracted (plasmid mini-prep kit, Tiangen, China) and electro-transferred into recipient strain, *E. coli* JM109SGL for various assays.

A LB composed of 5 g L$^{-1}$ yeast extract, 10 g L$^{-1}$ tryptone, and 10 g L$^{-1}$ NaCl was used for cell cultivation with antibiotics whatever necessary unless specifically noted. Electro-transferred cells were spread and grown on LB agar plate (LB media supplemented with 20 g L$^{-1}$ agar) with relevant antibiotics for obtaining positive colonies. Stock solution of 50 mg mL$^{-1}$ kanamycin (*Kan*) and 25 mg mL$^{-1}$ chloramphenicol (*Cm*) was prepared for uses throughout the cultivation processes of recombinant *E. coli* JM109SGL.

**Plasmids transformation**. For circuit characterization, plasmids of interest were electro-transformed into competent cells of *E. coli* JM109SGL. For electroporation, cells were first made electrocompetent by concentrating 100-folds and washing twice with ice-cold 10% glycerol and stored in −80 °C freezer before uses. Then, 50 μL of competent cells were mixed with 30–50 ng of the PCR products, and then electroporated at 1.8 kV with around 50 mA in an ice-cold 0.1 cm cuvette (Bio-Rad, USA), followed by the addition of 1 mL LB medium. After incubation at 37 °C for 1 h, cells were spread on agar plates with relevant antibiotics and grown for 12 h for colony selection via PCR analysis and sequencing. The positive colonies were selected for further study. For plasmid constructions, chemical competent *E. coli* DH5α were used to screen positive constructs of PCR assembly products. Five microliters of PCR products were mixed with 50 μL of competent cells after 30 min on ice. Followed by 45 s heat shock, cells were placed on ice for 2 min, then mixed with 0.95 mL fresh LB medium for 1 h incubation at 37 °C. Finally, 50 μL of incubated cells were spread on LB agar plate with relevant antibiotics and grown for 12 h for colony selections. All of the competent cells were stored at −80 °C before uses.

**Temperature-response function characterization**. To characterize the performance of various designs of T-switch (Supplementary File 9), all measurements of fluorescence intensity were taken by cytometer of cells in the beginning of stationary phase cultured in 96-deep-well plates. Glycerol stocks of the start host, *E. coli* JM109SGL, containing target plasmids were streaked and activated on LB agar plates for 12-h incubation at 37 °C. Single colonies were inoculated into 1 mL LB medium, followed by 12 h pre-culture in 2 mL well in 96-deep-well plates (NEST, China) sealed with an air permeable film (Axygen, USA) at 1000 rpm at different temperatures (30, 31, 32, 33, 34, 35, 36, and 37 °C), respectively (Thermal Shaker, AOSHENG, China). Then, pre-cultures were 200-fold diluted into 1 mL fresh LB medium for 12-h growth under the same conditions. After growth, cell cultures were 100-fold diluted into 250 μL PBS supplemented with 2 mg mL$^{-1}$ kanamycin to terminate the expression of proteins. Then, 0.1 vol% of 50 mg mL$^{-1}$ *Kan* and 25 mg mL$^{-1}$ *Cm* stock solution were added into the cultures for stabilizing plasmids of interest throughout the cultivation processes.

**On/off response during time course of cell growth (30–37 °C)**. To study the time course responses of T-switch designs, including the combinations of constructs 155+165, 147+167, and 163+166, recombinant *E. coli* JM109SGL harboring plasmids of interest were grown overnight on LB agar plates at 30 °C. After growth, single colonies were inoculated into 1 mL LB medium in 96-deep-well plates sealed with an air permeable film and grown for 12 h at 1000 rpm at 30 °C in a Thermal Shaker. Then, 5 μL of each culture was inoculated into 1 mL fresh LB medium for 12 h cultivation under the same conditions. Followed by 200-fold

dilution in 1 mL fresh LB medium, cells were grown for 12 h at 1000 rpm at 37 °C. During the growth, 2–10 μL of each culture was sampled in every 1 h and mixed with 250 μL of PBS supplemented with 2 mg mL$^{-1}$ *Kan*. Then, 0.1 vol% of *Kan* and *Cm* stock solution were added into the medium for stabilizing plasmids of interest throughout the cultivation processes. Fluorescence intensities of sfGFP and mRFP were measured by FACS.

**On/off response of dynamic control during different growth phases (30–37 °C).** The on/off performances of T-switch circuits in different growth phases were characterized by cytometer analysis. Single colonies of *E. coli* JM109SGL harboring plasmids of interest were grown on LB agar plates from glycerol stocks, and then inoculated into 1 mL LB medium for 12 h cultivation (1000 rpm, 30 °C, Thermal Shaker). After growth, 5 μL of each culture was transferred into 1 mL fresh LB media as inoculums. After 12-h growth under the same conditions, 5 μL cultures were diluted into 1 mL fresh LB medium and grown at 30 °C at 1000 rpm for 0, 2, 4, 6, 8, 10, and 12 h, respectively, then cultures were transferred to new shaker at 37 °C at 1000 rpm for a 12-h cultivation. Finally, cell cultures were 100-fold diluted into 250 μL PBS supplemented with 2 mg mL$^{-1}$ *Kan*. Then, 0.1 vol% of *Kan*, and *Cm* stock solution were added into the medium for stabilizing plasmids of interest throughout the cultural processes. Fluorescence intensities of sfGFP and mRFP were measured by FACS.

**Single-cell online recording on thermal responsive *E. coli*.** To online record the thermally responsive performances of engineered *E. coli* harboring T-switch of constructs 155+165, recombinant cells were incubated underneath a layer of solid 1.0% LB agarose (~1.5 mm) containing relevant antibiotics by placing 1 μL overnight incubated cell culture between a glass coverslip-bottomed 35 mm Petri dish with a glass diameter of 20 mm (Cellvis, USA). Cells were imaged on a Nikon A1RSi laser scanning confocal microscope equipped with a 100× (NA 1.40) oil-immersion lens. Images were obtained every 10 min and processed by NIS elements v4.60 in the end. Cells were maintained at required temperature during imaging with an active-control environmental chamber.

**Cytometry analysis.** The cytometry analysis was carried out by a BD LSR Fortessa flow cytometer with HTS attachment (BD, USA). Fluorescence positive cells were captured under the excitation spectrum of 488 nm (FITC channel, 440 V, sfGFP) and 584 nm (PE-Texas Red channel, 580 V, mRFP), the channels of forward scatter (FSC, 440 V) and side scatter (SSC, 260 V). Furthermore, cells were first gated by FSC and SSC (varied by different temperature) to illuminate noise events. Subsequently, fluorescence positive events were determined by fluorescence channels of FITC and Texa-red, respectively, to remove the fluorescence negative cells. Finally, cytometer data were processed and analyzed by FlowJo software (v10.7) for generating the mean value of fluorescence intensity. All source data were modified by the subtraction of fluorescence levels of negative control groups, which were *E. coli* JM109SGL harboring null vectors, the constructs 196+197.

**Tree ring-like colony assays on agar plates.** Assays were conducted on LB agar plates to generate tree ring-like colonies shown in Supplementary Figs. 9–11, respectively. Cell stocks of *E. coli* JM109SGL harboring T-switch circuits, including constructs 155+165, 147+167, and 147+168 groups, were streaked on LB agar plates and grown at 37 °C. After 12-h growth, single colony was picked and all were grown in a LB medium with relevant antibiotics for 12-h growth. Fifty microliters of cells cultures were spread on LB agar plate for *x* h (*x* = 6, 8, or 10) incubation, respectively, at 37 °C. Then the plates were moved to another incubator at 30 °C grown for (24–*x*) h, respectively. Finally, single-colony imaging was performed to generate tree ring-like patterns after three-time repetitive operations. Colonies of 155+165 group were recorded every 24 h of growth for tree ring-like formation analysis. Then, 0.1 vol% of *Kan* and *Cm* stock solution were added into the medium for stabilizing plasmids of interest throughout the assays.

Single colonies were photographed using Olympus SZX16 camera under Olympus U-RFL-T mercury lamp light combined with different filters (GFP: 436/20 EX filter and 480/40 EM filter; RFP: 572/35 EX filter and 645/75 EM filter). Images were taken at an aperture of f/2.8, and range of exposure times were typically between 0.01 and 5 s customized by optimizing the dynamic range of fluorescence of sfGFP and mRFP. Meanwhile, bright-field images of all colonies were photographed under ambient light exposure for comparative analysis. Photos were processed and adjusted by ImageJ (NIH, USA) with the size of 3.3 × 3.3 mm to generate figures.

**Thermal control of cell morphology in cell growth.** For static control of cell morphology by T-switch, single colony of recombinant *E. coli* JM109SGL harboring double plasmids 145+221 was inoculated into 1 mL LB medium and grown 12 h in a 96-deep-well plate (1000 rpm, Thermal Shaker) at different temperatures (30, 33, 35, and 37 °C, respectively). Subsequently, 2 μL of cell cultures were sampled for cell shape recording by bright-field imaging using Nikon Eclipse Ti-E inverted fluorescence microscope after appropriate dilution to satisfy the single-cell analysis.

For dynamic control of cell shape, the recombinant *E. coli* used above was first grown at 30 °C and 33 °C, respectively, in 1 mL LB medium for 12 h (1000 rpm,

Thermal Shaker). Then, cell cultures were injected into a microfluidic chip for single-cell capturing in specific chamber. Captured cells were grown at 30 °C initially, and temperature was changed to 33 °C in 1.5 h. Followed by 2.5-h growth, the temperature was modulated to 35 °C. Cell growth during the time course was recorded by bright-field imaging every 5 min. In contrast, captured cells from both 30 and 33 °C were grown at 35 °C constantly as comparative groups to study the thermal responsive morphology change in time course. Imaged with microscopy to measure cell shapes and cell lengths.

Cell growth imaging were generated using a Nikon Eclipse Ti-E inverted fluorescence microscope equipped with a 100× (NA 1.40) oil-immersion lens, together with an Andor DU885 EMCCD and/or Neo 5.5 sCMOS camera (Andor Technology, USA). Cells were maintained at needed temperature during imaging with an active-control environmental chamber. Images were collected using μManager v1.446 software to generate videos, then NIS elements v4.60 was used for adjusting the video. Cell lengths were measured by ImageJ manually. All assays are shown in Fig. 4 and Supplementary materials.

**Debugging of PHB and P4HB synthesis pathways in shake flask studies.** The recombinant cells of *E. coli* JM109SGL harboring construct 434 (PHB) and 435+68 (P4HB), respectively, were pre-cultured in 15 mL tubes containing 5 mL LB for 12 h at 200 rpm at 37 °C. Then seed cultures were 5 vol% inoculated into 100 mL LB medium supplemented with 20 g L$^{-1}$ glucose for 24 h cultivation. Then, 0.1 vol% of *Kan* and/or *Cm* stock solution were added into the medium for stabilizing plasmids of interest throughout the cultural processes. IPTG was added after 4 h cultivation whatever necessary. Then, 30 mL of fermentation broths were sampled for further analysis.

**Fed-batch studies for copolymer productions of 3HB and 4HB, namely, PHB-*b*-P4HB and P(3HB-*co*-4HB).** Single colonies of recombinant cells containing target plasmids, constructs 169+170 for PHB-*b*-P4HB production and 149+150 for P(3HB-*co*-4HB) production, were grown in 5 mL LB medium as inoculums for 12 h at 30 °C at 200 rpm. Then, these pre-cultures were 1 vol% inoculated into 500 mL shake flasks carrying 100 mL LB medium and grown under the same condition for 12 h for seed culture preparation. Subsequently, 300 mL of seed cultures were inoculated into 2.7 L LB medium supplemented with 20 g L$^{-1}$ glucose. All fed-batch studies were carried out for 32 h. Agitation speed was coupled with dissolved oxygen (~30%) to reach 800 rpm from 200 rpm during the fermentation process. Particularly, 100 mL of feeding solution I, Feed-I (g L$^{-1}$) containing 300 glucose, 22.5 NH$_4$Cl, and 80 Yeast Extract, was added in a feeding rate of 0.5 mL min$^{-1}$ when the residual glucose decreased to lower than 5 g L$^{-1}$. Once Feed-I exhausted, the feeding solution II, Feed-II (g L$^{-1}$) consisting of 600 glucose, should be added to maintain the concentration of residual glucose at 50–70 mmol L$^{-1}$ (9–12.6 g L$^{-1}$) by real-time monitoring of glucose measured using Clinistix paper (SANNUO, China). The pH was automatically adjusted at 7.0 using 5 mol L$^{-1}$ NaOH solution. Then, 20 mL of fermentation broth was sampled for further analysis during the bioprocessing whatever necessary.

**Assays of dry cell mass and PHA contents.** Cell growth and PHA content analyses were carried out to study the dynamic performances of copolymer production controlled by T-switch during fed-batch fermentations. First, 20 or 30 mL of fermentation broth of each time point was harvested by centrifugation at 12,000 rpm for 10 min, and washed by distilled water twice. Then, the bacterial precipitates were freeze-dried using a vacuum lyophilizer (LGJ-10 C, Beijing Sihuan, China) under −80 °C, and weighted for calculating the cell dry weights (CDW, g L$^{-1}$). Second, 20–40 mg lyophilized cells in powder forms were sampled for methanolysis in a 15-mL tube containing 2 mL chloroform and 2 mL methanolysis solution (85 wt% methanol, 15 wt% H$_2$SO$_4$, and 1 g L$^{-1}$ benzonic acid), followed by extraction mixed with 1 mL distilled water in the cooled methanolysis solution, approximately 1 mL of extract liquor was used for gas chromatography analysis (GC-2014, SHIMADZU, Japan) to measure PHA content and monomer compositions[36]. About 10 mg of PHB standard (Sigma-Aldrich) and 10 mg of γ-butyrolactone (4HB standard, Sigma-Aldrich) were used as standards here.

**NMR and D-value.** 3HB and 4HB copolymers were extracted using chloroform solvent in a Soxhlet extractor (Soxtec 2050, Foss, Denmark) from lyophilized cells harvested from fermentation broth. Subsequently, the extracted PHA was dissolved in chloroform and then precipitated with ten-fold volume of ethanol. After centrifugation at 12,000 rpm for 10 min, the resulted PHA was oven dried at 65 °C for 12 h prior to the subsequent studies. All fractionated polymers were analyzed by $^{13}$C NMR (Nuclear Magnetic Resonance) (Oxford-600, UK) for identifying block copolymer and/or random copolymer of 3HB and 4HB[49]. MestReNova12 (Mestrelab Research, Spain) was used for spectra analysis. Bernoullian statistics method[50] was employed to calculate the D-value of block copolymer and random copolymer (Supplementary Figs. 17, 19, and 21).

**DSC analysis.** DSC was used to measure crystallinity degree, melting temperature, and glass-transition temperature of PHB, P4HB, and their copolymers, respectively, via a TA Instrument (DSC-Q20, TA, USA). A sample of 3–10 mg was compressed

in an aluminum-sealed pan. Then the pan was cooled to −80 °C, then heated from −80 to 180 °C at a rate of 10 °C min$^{-1}$. The sample was maintained at 180 °C for 2 min under a nitrogen atmosphere of 50 mL min$^{-1}$. Then, the pan was quenched to −80 °C and reheated from −80 to 180 °C at a rate of 10 °C min$^{-1}$. Data were collected during the second heating run[49]. P4HB purchased from Tepha (USA) and PHB from Sigma-Aldrich were used as standards for comparative analysis.

**Figures generation**. Figures were generated through Adobe Illustrator CC2017, Prism v8 (GraphPad), Microsoft Office 2016 (Power Point, Excel, and Word), and ImageJ whatever necessary. Figures of temperature-response performance on LB agar plates were generated by using color palettes of ImageJ. FI of agar plate experiments were normalized by the maximum FI value of sfGFP and mRFP measured from every single image, ranging from 0 to 100.

**Reporting summary**. Further information on research design is available in the Nature Research Reporting Summary linked to this article.

## Data availability
The authors declare that source data processed for figure generation in this study are available within the paper and its Supplementary Information files. Plasmids used in this study, OD600 measured in fermentation experiments are deposited in Source Data file. The datasets generated and analyzed during the current study are available from the corresponding authors upon request. Any other relevant data are available from the authors upon reasonable request. Source Data are provided with this paper.

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

## Acknowledgements

This research was financially supported by grants from the Ministry of Science and Technology of China (Grant No. 2018YFA0900200; No. 2016YFB0302500), National Natural Science Foundation of China (Grant No. 21761132013; No. 31870859; No. 32001029), and Tsinghua University-INDITEX Sustainable Development Fund (Grant No. TISD201907). This project is also funded by the National Natural Science Foundation of China (Grant No. 31961133017, No. 31961133018, No. 31961133019). These grants are part of MIX-UP, a joint NSFC and EU H2020 collaboration. In Europe, MIX-UP has received funding from the European Union's Horizon 2020 research and innovation program under grant agreement No. 870294. The PhlF repressor encoded gene was donated by Professor Chunbo Lou from SIAT in Shenzhen, China. The PHA mechanical property analysis was carried out with the help of Professor Jun Xu from Department of Chemical Engineering, Tsinghua University.

## Author contributions

G.-Q.C. and J.-W.Y. proposed the idea. G.-Q.C. and J.-W.Y. designed and supervised the experiments. J.-W.Y. performed the prototype of T-switch construction. X.W. and J.-N.H. performed the characterization and optimization experiments of T-switch, as well as the constructs for copolymers biosynthesis and morphology manipulation. X.W., J.-N.H., and J.-W.Y. performed fed-batch fermentation for block- and random-copolymers. X.W., J.-N.H. and X.Z. performed the materials analysis. X.W., J.-W.Y., and D.-J.L. performed the single cells monitoring experiments. X.W., J.-N.H., Y.-Y.M., Y.L., H.W. and T.-R.Z. performed all other experiments. J.-W.Y., X.W., and J.-N.H. analyzed the data and wrote the manuscript. G.-Q.C, J.-W.Y., and F.-Q.W. revised the manuscript. All authors have read and approved the final manuscript.

## Competing interests

The authors declare no competing interests.
