## [Peer Review File · Nature Communications]

Reviewers' Comments:

Reviewer #1:

Remarks to the Author:

The article "Thermal regulation for bifunctional dynamic control of gene expression in *Escherichia coli*" by Wang and collaborators describes a genetic circuit, based on a thermosensitive switch that enabled bidirectional control of gene expression over time. The synthetic circuit is used to demonstrate morphology changes in living *E. coli* cells to achieve tree ring-like bacterial colonies with hierarchical patterns and also to switch cells between spherical-, rod- and fiber-shape by modifying the temperature. A metabolic engineering application is further demonstrated by adopting the circuit to control the levels of 3-hydroxybutyrate and 4-hydroxybutyrate as components of polyhydroxyalkanoate inclusions in engineered *E. coli*. This is a very solid article, with a clear structure. The work is a nice demonstration of the benefits of using synthetic circuits to control production pathways, and the results are interesting (but perhaps not too surprising). The manuscript could benefit from addressing the following points, in no specific order of importance:

(1) The use of the English language should be thoroughly checked in the manuscript. In particular, the English could use editing in several places for subject-verb agreement, correct use of adjectives and adverbs, etc.

(2) Prior to publication, I would recommend the authors to include full sequences of all relevant genes and parts in this study in the supplemental material. The SynBio community agrees that having this sort of information transparently present in the supplemental is critical for enabling use of the technologies by different scientists. Note that Table S2 includes part of this information, but this should be expanded (also, fix 'primers' in this supplemental table).

(3) L183-184: How was this cultivation regime selected? Did the authors optimize time and temperature as the variables for this experiment?

(4) All the experiments (both for synthetic circuit design and bioproduction) seem to have been carried out in LB medium: what is the influence of a complex culture medium on the circuit performance? The authors should show how the system works in minimal media, with a defined carbon source instead of a complex mixture of nutrients.

(5) What is the expected impact of temperature heterogeneities (say, for instance, in a large bioreactor) on the circuit performance (and thus on the expected levels of polymer accumulation)? The authors could discuss how this parameter could influence the overall output in the engineered strains.

(6) In connection with the point above, did the authors test the engineered cell's behavior when they are cultivated at an intermediate temperature (e.g. 35°C)? This type of experiment would help understanding the dynamics (and probably, stochasticity) of the system.

(7) A thorough statistical analysis is needed in Figs. 3d, 3e, 4c, and 4d. Please include the relevant P values in the figure panels.

Reviewer #2:

Remarks to the Author:

In this work, the authors leverage temperature sensitive cI857 to construct a "T-switch" to create thermal regulation of gene expression. This system is used to drive expression of at least two gene cassettes simultaneously in opposing directions via an inverter circuit (repressor protein in one

cassette that represses the other cassette) creating bifunctionality. The authors fine tune this response somewhat via the addition of degradation tags and introducing negative feedback to increase system ultrasensitivity and reduce leakiness. Using these systems, the authors induce pattern formation in *E. coli* colonies (tree ring-like structures with different fluorescent reporters), cellular morphology (rods to spheres or fibres), and in creating PHA block copolymers via temperature cycling. While the data is fairly robust, complete, and sufficient to support the conclusions, the writing is wholly opaque making it difficult to discern the novel contributions of the authors. First, the novel contribution here is the demonstration of spatial patterning in cellular behavior via reversible thermal control. This should be reflected in the paper title and the introduction restructured to match rather than emphasizing gene regulation or control as the system (cI857) is well established. Similarly, the results are focused on many procedural details and do not fully discuss the rationale of design choices or support them with models or prior literature making it difficult for the reader to fully interpret the results. E.g. why is negative feedback needed? Was this really needed to optimize your design? While the manuscript may be of interest to readers, it would first benefit from a significant rewrite to improve clarity.

(Just a few) Specific comments:

Lines 40-41: "This study demonstrates the possibility of well-organized chemosynthesis-like manner" It seems a word or two are missing here as I have no idea what this is trying to say.

Lines 50-51: "artificial gene circuits of scalability and robustness" -> "scalable and robust gene circuits"

Replicates and variability (error bars) are unclear for Figure 1,

Line 249: "to our knowledge" is not needed

Reviewer #3:

Remarks to the Author:

The authors describe a design of genetic circuits that contain a thermo-switch. They have presented a series of optimization to obtain tunable genetic circuits with low leakage. Four different output modules are incorporated into the circuits to evaluate the tunability of this thermo-responsive regulation.

In this study, the results based on four different circuits have clearly verified the programmability and tunability of the thermo-switch system. The authors employed additional genetic elements to achieve tighter control of outputs via the thermo-switch system. However, the selection of PhIF as the key component is dubious as transcriptome profiling demonstrated as many as 250 genes were significantly down-regulated at 37°C in the bacterial host used in this study. It is evident that the output was not affected by this down-regulation at 37°C, but the tunability of the thermo-switch might be effective if other output modules are used. A "neutral" modulator is desired to enable the system more universal and compatible.

For the three ring-like colony assays, the authors did not discuss the inequivalent expression patterns of the red and green reporter genes. Why they are expressed in different patterns (intensity, band width, different number of rings/peak value)? This should be addressed in a revision since it implies the inherent characteristics of the genetic circuits and the reporters.

Overall, the design and results of this work is sound and solid. The English of this manuscript needs to be professionally edited to make it more readable. For example, "Genetically programming circuits allowing simultaneous up- and down-regulation of enzyme expression has far-reaching utilities for many bio-manufacturing purposes." How can an enzyme be simultaneously

up- and down-regulated? I guess that the authors intended to say that two enzymes can be up- and down-regulated by the same genetic circuit at the same time

Listed Responses to Reviewers of Ref: **NCOMMS-20-45493-T**

Title: **Reversible thermal regulation for bifunctional dynamic control of gene expression in *Escherichia coli***

We are grateful to the editor and the **three reviewers** for their efforts to handle and review our paper. After carefully studying the comments and suggestions, following responses have been generated (highlighted in **red** color):

Comments from reviewers:

Reviewer #1 (Remarks to the Author):

The article “Thermal regulation for bifunctional dynamic control of gene expression in *Escherichia coli*” by Wang and collaborators describes a genetic circuit, based on a thermosensitive switch that enabled bidirectional control of gene expression over time. The synthetic circuit is used to demonstrate morphology changes in living *E. coli* cells to achieve tree ring-like bacterial colonies with hierarchical patterns and also to switch cells between spherical-, rod- and fiber-shape by modifying the temperature. A metabolic engineering application is further demonstrated by adopting the circuit to control the levels of 3-hydroxybutyrate and 4-hydroxybutyrate as components of polyhydroxyalkanoate inclusions in engineered *E. coli*. This is a very solid article, with a clear structure. The work is a nice demonstration of the benefits of using synthetic circuits to control production pathways, and the results are interesting (but perhaps not too surprising). The manuscript could benefit from addressing the following points, in no specific order of importance:

Response: Thank you for your very good summary of our study.

(1) The use of the English language should be thoroughly checked in the manuscript. In particular, the English could use editing in several places for subject-verb agreement, correct use of adjectives and adverbs, etc.

Response: Thank you for your suggestion. The manuscript has been thoroughly revised to correct the use of adjectives and adverbs.

(2) Prior to publication, I would recommend the authors to include full sequences of all relevant genes and parts in this study in the supplemental material. The SynBio community agrees that having this sort of information transparently present in the supplemental is critical for enabling use of the technologies by different scientists.

Response: Thank you for your constructive suggestions. We have provided full sequences of all relevant constructs in our study, including vectors, genes, parts and other detailed information that support the reuses by different scientists, within the **Plasmids folder of **Source Data.Zip** file (see**

the screen shot below). Descriptions of each part are in **Source Data.xlsx – Parts Description Sheet**.

Note that Table S2 includes part of this information, but this should be expanded (also, fix ‘primers’ in this supplemental table).

Response: Thanks and done! “Primers” have been fixed.

(3) L183-184: How was this cultivation regime selected? Did the authors optimize time and temperature as the variables for this experiment?

Response: In this experiment, we implemented three different incubation strategies: I) 37° 10 h and 30° 14 h, II) 37° 8 h and 30° 16 h, II) 37° 6 h and 30° 18 h, per day as a cycle, the incubation time at 37° was set as X (X = 10, 8 and 6); three cycles in total (72 h). The results are displayed in **Supplementary Figures 9-11**. The firstly generated red ring (R1) could only be observed when X=10, which implies that 6 and 8 h incubation at 30° (X = 8 and 6) was too short to form the initial red ring of the colonies. As a result, X = 10 was chosen to obtain distinct double-color rings as a cultivation regime here.

(4) All the experiments (both for synthetic circuit design and bioproduction) seem to have been carried out in LB medium: what is the influence of a complex culture medium on the circuit performance? The authors should show how the system works in minimal media, with a defined carbon source instead of a complex mixture of nutrients.

Response: Thank you for your constructive suggestions. Based on your advices, we accessed the performances of our primary circuit (construct 155+165) in LB and M9 media, respectively, at 30, 33, 37°, the three typical temperatures with basal and/or saturated outputs, and the turnover points of these two phases (see **Figure 1d**). The results were shown in **Supplementary Figure 2 (Culture components and culture methods were shown in Supplementary Information)**, also attached below, which shows similar expression control performances of mRFP and sfGFP in M9 and LB media, respectively. Source data are provided in the **Source Data** file (*.xlsx).

Supplementary Figure 2 Comparative analysis of T-switch performance in LB and M9 medium.

The recombinant cells harboring T-switch (construct 155+165) grown in the LB and M9 media, respectively, at 30, 33 and 37°C, were harvested for fluorescence measurement of sfGFP (a) or mRFP (b) by FACS (see methods). Error bars, mean \pm s.d. of three replicates (data points in circle). FI, Fluorescence Intensity in arbitrary unit (a.u.).

(5) What is the expected impact of temperature heterogeneities (say, for instance, in a large bioreactor) on the circuit performance (and thus on the expected levels of polymer accumulation)? The authors could discuss how this parameter could influence the overall output in the engineered strains.

Response: The temperature heterogeneities in a large bioreactor could be significant (generally with control bias no greater than 0.5 °C when the T becomes stable), because the heat transfer parameters are well determined based on the theoretical calculation. However, the temperature alteration in larger bioreactor consumes longer time to achieve stable control at a specific temperature, which may lead to a lag response-control performance in industry. Thus, tuning the operational range of temperature can not only achieve fast response for temperature control, but also alleviate the heat shock effects on cellular activities. Generally, the polymer accumulation levels are not affected by cultural scale in different bioreactors using an optimized fed-batch process design, these have been demonstrated successfully in scale-up (from 1 L to 5000 L bioreactor) for PHA productions. Meanwhile, the production titer of polymers in this study had a lot of spaces for improvement using static optimization strategy and feeding strategy during fed-batch cultivations, these were reported in previous studies.

(6) In connection with the point above, did the authors test the engineered cell's behavior when they are cultivated at an intermediate temperature (e.g. 35°C)? This type of experiment would help understanding the dynamics (and probably, stochasticity) of the system.

Response: Thank you for your good question. For circuit characterization using reporters (such as mRFP and sfGFP), we tested the bifunctional control behavior of the engineered cells at different intermediate temperatures from 30 to 37 °C (Figs. 1c-e, and Sup Figs 3 and 8) in the LB medium, respectively. Additionally, we also studied the performances of construct 155+165 in the M9

medium at 30, 33 and 37 °C, respectively. Since 33 or 35 °C is an important intermediate temperature representing the turnover point of the bidirectional control phases (**Fig. 1d**).

For case studies, the morphology of engineered *E. coli* harboring TM-switch cultured at 33°C exhibited negligible variance compared with the start host (*E. coli* JM109sgl), which is consistent with the characterization results mentioned above. Besides, the cell morphology behavior at 30, 35 and 37 °C were also studied, respectively. However, the recombinant cells grown at 37°C were negatively affected probably due to the strong over-expression of *ftsZ* gene, which is discussed in the revised MS. Practically, the tests of the colony ring formation and block copolymer synthesis at different intermediate temperatures were not included in this study, because stringent expression control at 30 and 37 °C was prior for pattern formation whatever on macroscopic or molecular scales. However, the random copolymer synthesis assays conducted in bioreactors were studied at 30, 31 and 32 °C, respectively, to generate P(3HB-co-4HB) with different 4HB monomer fraction by modulating the growth temperature.

In summary, the engineered cells' behavior cultivated at different temperatures was well characterized to enable the rational design of case studies in various applications.

(7) A thorough statistical analysis is needed in Figs. 3d, 3e, 4c, and 4d. Please include the relevant P values in the figure panels.

Response: Statistical analysis of **Figures 3d-3e** were partially plotted in **Supplementary Fig 12** independently. And all of the t-test calculation results were included in **Source Data.xlsx** file.

Supplementary Figure 12 Growth effects on tree ring-like colony formation.

(a) Diameter of tree ring-like colonies increased with longer temperature cycles at 37°C (X). (b) Ring width of both sfGFP and mRFP was significantly affected by the colony growth rate (log phase and stationary phase with p value horizontally plotting, X = 10 h) and incubation time X (p value plotted vertically, X = 6 and 10 h). Hollow and solid squares represent the visible and missing formation of the first red ring (R1) generated by constructs 147+167, respectively, namely 147+167-D (2 colonies) and 147+167-S (4 colonies), respectively. Error bars in d and e, mean ± s.d. of at least 2 collected colonies. p values were partially shown in this figure due to the limited space, all of the one-way ANOVA calculation results are included in Source Data file; N.S. not significant; * p < 0.0332; ** p < 0.0021; *** p < 0.0002; and **** p < 0.0001.

Statistical analysis is added in **Figs. 4c** and **4d** with relevant P values described in legends. Specifically, the cell lengths from **Figs. 4c-4d** display a log-normal distribution (Kaya and Koser 2009), and t-test analysis was employed to determine the P values from **Figs. 4c-4d** (also showed below).

Fig. 4c

Fig. 4d

Figure 4 Thermal responsive cell morphologies changes from rod to spheres or to fibers

(c) Quantitative measurements of cell lengths by imageJ from at least 9 images from part b containing over 150 captured cells manually. (d) Bifunctional dynamic control of cell morphology among shapes of spheres (mreB overexpressing), rods (normal cell type) and fibers (ftsZ overexpressing) were on-line recorded in a microfluidics with a scale bar of 2 μm . Sample sizes of collected cells of each time point varied significantly depending on the growth phase, 3-10 cells at 1 h, 10-40 cells at 3 h, and 100-240 cells at 5 h. All data from experiments c and d are displayed in Box-plot a value of median, quantiles, mini- and maxi-mum. p value: N.S. not significant; * $p < 0.0332$; ** $p < 0.0021$; *** $p < 0.0002$; and **** $p < 0.0001$.

Reference: Kaya, Tolga , and H. Koser . Characterization of Hydrodynamic Surface Interactions of *Escherichia coli* Cell Bodies in Shear Flow. *Physical Review Letters* 103.13(2009):138103.

Reviewer #2 (Remarks to the Author):

In this work, the authors leverage temperature sensitive cI857 to construct a “T-switch” to create thermal regulation of gene expression. This system is used to drive expression of at least two gene cassettes simultaneously in opposing directions via an inverter circuit (repressor protein in one cassette that represses the other cassette) creating bifunctionality. The authors fine tune this response somewhat via the addition of degradation tags and introducing negative feedback to increase system ultrasensitivity and reduce leakiness. Using these systems, the authors induce pattern formation in *E. coli* colonies (tree ring-like structures with different fluorescent reporters), cellular morphology (rods to spheres or fibres), and in creating PHA block copolymers via temperature cycling. While the data is fairly robust, complete, and sufficient to support the conclusions, the writing is wholly opaque making it difficult to discern the novel contributions of the authors. First, the novel contribution here is the demonstration of spatial patterning in cellular behavior via reversible thermal control. This should be reflected in the paper title and the introduction restructured to match rather than emphasizing gene regulation or control as the system (cI857) is well established. Similarly, the results are focused on many procedural details and do not fully discuss the rationale of design choices or support them with models or prior literature making it difficult for the reader to fully interpret the results. E.g. why is negative feedback needed? Was this really needed to optimize your design? While the manuscript may be of interest to readers, it would first benefit from a significant rewrite to improve clarity.

Response: Thanks for your comprehensive summary, nice comments and good suggestions. The manuscript has been thoroughly revised and highlighted in the MS with color. The demonstration of **‘spatial patterning in cellular behavior via reversible thermal control’** is now emphasized in both ‘abstract’ and ‘results’ section. The title has been changed to “Reversible thermal regulation for bifunctional dynamic control of gene expression in *Escherichia coli*” to better reflect this study.

”

‘why is negative feedback needed? Was this really needed to optimize your design?’

Response: The negative feedback control is necessary for distinct red ring formation to avoid reversible activation of mRFP expression from 30 to 37°C (see the cartoon video in Supplementary files) when compared to stepwise patterns by construct 155+165. Detailed explanation has been added in the revised MS.

(Just a few) Specific comments:

Lines 40-41: “This study demonstrates the possibility of well-organized chemosynthesis-like manner” It seems a word or two are missing here as I have no idea what this is trying to say.

Response: Thank you. This sentence has been revised as “This study demonstrates the possibility of well-organized, chemosynthesis-like block polymers on a molecular scale in reprogrammed microbes”.

Lines 50-51: “artificial gene circuits of scalability and robustness” -> “scalable and robust gene circuits”

Response: Thank you and done.

Replicates and variability (error bars) are unclear for Figure 1,

Response: Three replicates were used for experiments from **Figure 1** as described in its legend. The variability of the data in **Figure 1** was too small to be displayed in some data points, thus as the software Prism claims: “If the error bar would be shorter than the size of the symbol, Prism simply won't draw it, even if the symbol is clear”. However, all of the relevant source data has been provided in **Source Data.xlsx – Fig 1c-e** sheet.

Line 249: “to our knowledge” is not needed

Response: Thanks. The “to our knowledge” has been removed.

Reviewer #3 (Remarks to the Author):

The authors describe a design of genetic circuits that contain a thermo-switch. They have presented a series of optimizations to obtain tunable genetic circuits with low leakage. Four different output modules are incorporated into the circuits to evaluate the tunability of this thermo-responsive regulation.

Response: Thanks for your precise summary of our study.

In this study, the results based on four different circuits have clearly verified the programmability and tunability of the thermo-switch system. The authors employed additional genetic elements to achieve tighter control of outputs via the thermo-switch system. However, the selection of PhIF as the key component is dubious as transcriptome profiling demonstrated as many as 250 genes were significantly down-regulated at 37°C in the bacterial host used in this study. It is evident that the output was not affected by this down-regulation at 37°C, but the tunability of the thermo-switch might be effective if other output modules are used. A “neutral” modulator is desired to enable the system more universal and compatible.

Response: Thank you for constructive comments. In this study, the 4HB (4-hydroxybutyrate) block synthesis pathway is very sensitive to low expression levels with a saturated output in the presence of 2 mg/L IPTG (**Figure 5b**). Thus, ‘PhIF’ is still a good choice for the block copolymer synthesis due to its tight control performance on P_{PhIF} promoter (ultra-low basal expression), which other regulator systems cannot easily achieve without heavy engineering efforts. However, based on the transcriptome profiling, the ‘PhIF’ repressor does influence cell activity, indicating that a ‘neutral’ modulator, such as LacI, is more suitable for universal and compatible uses except P(3HB-co-4HB) synthesis and other metabolic pathways with a low expression level-response activity. This is what we are planning to develop in the coming study.

For the three ring-like colony assays, the authors did not discuss the inequivalent expression patterns of the red and green reporter genes. Why they are expressed in different patterns

(intensity, band width, different number of rings/peak value)? This should be addressed in a revision since it implies the inherent characteristics of the genetic circuits and the reporters.

Response: Thanks for your good advices. We have listed two separated responses (I and II) to address your comments. **I)** ‘inequivalent expression patterns of the red and green reporter genes’ For the construct 155+165, the sfGFP rings were clearly formed with visible boundary due to the durable repression of PhIF once the colony was incubated at 37°C. However, the mRFP signal showed step-wise enhanced tendency from outer- to inner-region, because the *in-situ* memorial expression of mRFP was periodically activated once the colony was exposed at 37°C (see cartoon video in **Supplementary File 2**). Based on the success of green rings formation, we then introduced a negative feedback control modulator (LacI-LacO system, construct 147+167) to inhibit the recovered expression of mRFP from 30 to 37°C, achieving distinct red rings from the stepwise pattern. Interestingly, the addition of degradation tag in the C-terminal of mRFP can further improve the red ring formation (**Figure 3 and Supplementary Figures 10, 11**).

II) ‘Why they are expressed in different patterns (intensity, band width, different number of rings/peak value)?’

First, both of the red and green rings exhibit decreased fluorescence intensity (peak values) from the inner- to outer-ring, which is relevant to the growth phase of colony formation. Because the protein expression activity decreases while colony growth is turning to the stationary phase.

Second, the band width of color rings is affected by two factors: a) The incubation time X at 37°C (the same as 24-X at 30°C). Generally, a larger band width of red rings would be formed with an increased X, so does the green rings with an increase 24-X; b) The growth phase of colony formation, similar explanation as described in the ‘First’ part.

Third, the transition region of green and red rings were enlarged with the introduction of the double repression system, PhIF-PhIO and LacI-LacO, as the recovered activation of relevant reporter was depended on the dilution of cell division to form a non-color gap of two neighbor rings.

Discussion of the above information has been revised in the manuscript together with a cartoon video (**Supplementary Video 2**) listed in Supplementary files for better understanding of this colony assays.

Overall, the design and results of this work is sound and solid. The English of this manuscript needs to be professionally edited to make it more readable. For example, “Genetically programming circuits allowing simultaneous up- and down-regulation of enzyme expression has far-reaching utilities for many bio-manufacturing purposes.” How can an enzyme be simultaneously up- and down-regulated? I guess that the authors intended to say that two enzymes can be up- and down-regulated by the same genetic circuit at the same time

Response: Thanks for your good understanding. The manuscript has been thoroughly revised.

We are very grateful for the constructive comments and advices from the three reviewers. Your inputs have helped significantly improved the quality of this paper.

Reviewers' Comments:

Reviewer #1:

Remarks to the Author:

Thanks for addressing all my comments and criticism. The manuscript has improved significantly and I only have some minor suggestions as follows:

- In several places of the text, it seems that the '°C' symbol is missing. This can be fixed at the proofing stage.
- Fig. 5a. Please draw an arrow connecting acetyl-CoA and the TCA cycle, as this metabolite is also used in the cycle and the current figure does not reflect this important aspect. In the same figure, please use the nomenclature for enzymes and not genes besides each reaction (i.e. SucD and not lowercase, italics *sucD*).

Reviewer #2:

Remarks to the Author:

Thank you for addressing our technical concerns and strengthening your arguments. However, there is still a lack of clarity in the manuscript with regards to scientific rationale. For example, my comment regarding the rationale for negative feedback and reviewer #3's comment on your choice of pHIF.

With the example of negative feedback, I assume negative feedback is used to reduce the leaky expression of the system as measured by mRFP levels. However, this is never explicitly stated nor is it explained why that leakiness is an issue. Does your system need to have a certain level of tightness? Why?

Reviewer #3:

Remarks to the Author:

The comments of the reviewers are well addressed. The wording and phrasing problem has improved but still needs further polishing. For example, L53, "mono-" should be "uni-"; L77-78, "engineering thermal-switchable bioswitch of bifunction is possible to offer a rich set of tools for simultaneously activating ..." can be rephrased as " engineering thermal-switchable bioswitch of bifunction can make possible the simultaneous activation ..."; L193-196, "Notably, the sfGFP rings were distinguishable while the mRFP fluorescent intensity (FI) was stepwise enhanced from outer- to inner-region. Because the in-situ memorial expression of mRFP was periodically activated and stacked once the colony was exposed at 37-." Are these sentences separate ones or should be merged into one? L220-221, "since the recovered expression of related reporter was highly depended on the radial dilution effect of cell division." "was highly depended on " should be revised as "highly depends on". Lastly, in the section "Tree ring-like patterning by temperature-response grown colonies", some hypothesis/explanation can be moved to discussion section. The discussion about this section could be expanded in the Discussion section.

Listed Responses to Reviewers of Ref: **NCOMMS-20-45493A**

Title: **Reversible thermal regulation for bifunctional dynamic control of gene expression in *Escherichia coli***

We are grateful to the editor and the **three reviewers** for their efforts to handle and review our paper. After carefully studying the comments and suggestions, following responses have been generated (highlighted in red color):

Comments from reviewers:

Reviewer #1 (Remarks to the Author):

Thanks for addressing all my comments and criticism. The manuscript has improved significantly

Response: Thank you for your kindly comments and criticism to improve the manuscript.

I only have some minor suggestions as follows: In several places of the text, it seems that the "°C" symbol is missing. This can be fixed at the proofing stage.

Response: Thank you for your suggestion. The manuscript has been thoroughly revised to add the "°C" symbol.

- Fig. 5a. Please draw an arrow connecting acetyl-CoA and the TCA cycle, as this metabolite is also used in the cycle and the current figure does not reflect this important aspect. In the same figure, please use the nomenclature for enzymes and not genes besides each reaction (i.e. SucD and not lowercase, italics *sucD*).

Response: Thank you for your suggestion. An arrow connecting acetyl-CoA and the TCA cycle was added in Figure 5a, as well as the genes in lowercase, italics forms were adjusted to the nomenclature for enzyme, such as SucD, 4HBD, PhaC, PhaA, PhaB.

Reviewer #2 (Remarks to the Author):

Thank you for addressing our technical concerns and strengthening your arguments.

Response: Thank you for your nice reply.

However, there is still a lack of clarity in the manuscript with regards to scientific rationale. For example, my comment regarding the rationale for negative feedback and reviewer #3's comment on your choice of *phIF*. With the example of negative feedback, I assume negative feedback is used to reduce the leaky expression of the system as measured by mRFP levels. However, this is never explicitly stated nor is it explained why that leakiness is an issue. Does your system need to have a certain level of tightness? Why?

Response: Thank you for your comments.

Here, we split two aspects to address your question:

1) **Negative control of mRFP using LacI repressor.**

It is important to note that the expression pattern of mRFP controlled by P_R promoter (direct temperature control panel) is totally different from sfGFP controlled by P_{PhIF} , because cells in the rings formed at 30°C previously are able to express mRFP in the later growth at 37°C with the absent of LacI, while sfGFP is tightly repressed once the PhIF expression occurs. Thus, the introduction of negative feedback control here aims to repress the recovered expression of mRFP when colony was exposure to 37°C periodically rather than reducing the leakiness. The schematic and experimental (from **Fig. 3c sheet** of source data file (.xlsx)) figures were co-organized and shown below to provide intuitive illustration in corporation of the supplementary video (**Supplementary File 2**) with independent pattern of sfGFP, mRFP and merged colony rings.

2) **The choice of PhIF.** Firstly, in our work, the 4HB (4-hydroxybutyrate) block synthesis pathway has proven ultra-sensitivity to low expression level showing a saturated output only in the presence of 2 mg/L IPTG, while the 3HB block synthesis pathway exhibited negligible activity under the induction of 2 mg/L IPTG, and PHB accumulation was significantly improved with higher supplemented dosage of IPTG higher than 2 mg/L IPTG, such as 20, 200 and 1000 showed in **Figure 5b**. These findings indicates that 4HB synthesis pathway should be tightly controlled using a repressor with ultra-low basal expression, such as PhIF, to fully reduce the leakage effects. However, the rigorous ON/OFF control of 3HB synthesis pathway is much easier to achieve due to its poor performance under low expression level. The data from **Figure 1c** (different leakiness level of two control panels), **Figure 5c** and **Supplementary Figure 15** (sensitivity assays of 3HB and 4HB pathway to low expression level), **Supplementary Figures 16a** (circuit design for **block** copolymer synthesis, 4HB pathway was control by P_{PhIF} promoter), as well as **Supplementary Figures 20** (circuit design for **random** copolymer synthesis, 4HB pathway was control by P_R promoter with a certain leakiness) can strongly explain the choice of PhIF.

Once again, thanks a lot and hope our answers can address your questions.

Reviewer #3 (Remarks to the Author):

The comments of the reviewers are well addressed.

Response: Thank you for your kindly review of our revised manuscript.

The wording and phrasing problem has improved but still needs further polishing.
For example, L53, “mono-“ should be “uni-“;

Response: Thank you for your suggestion. This word has been revised.

L77-78, “engineering thermal-switchable bioswitch of bifunction is possible to offer a rich set of tools for simultaneously activating ...” can be rephrased as “ engineering thermal switchable bioswitch of bifunction can make possible the simultaneous activation ...”;

Response: Thank you for your suggestion. This sentence has been revised.

L193-196, “Notably, the sfGFP rings were distinguishable while the mRFP fluorescent intensity (FI) was stepwise enhanced from outer- to inner-region. Because the in-situ memorial expression of mRFP was periodically activated and stacked once the colony was exposed at 37^o.” Are these sentences separate ones or should be merged into one?

Response: Thank you for your suggestion. These sentences should be merged into one.

L220-221, “since the recovered expression of related reporter was highly depended on the radial dilution effect of cell division.” “was highly depended on “should be revised as “highly depends on”. Lastly, in the section “Tree ring-like patterning by temperature-response grown colonies”, some hypothesis/explanation can be moved to discussion section. The discussion about this section could be expanded in the Discussion section.

Response: Thank you for your suggestion. This sentence has been revised.

Overall, thanks again for the nice comments from all reviewers to improve the readability and science of the manuscript.